# Chloroplast Envelopes Play a Role in the Formation of Autophagy-Related Structures in Plants

**DOI:** 10.3390/plants12030443

**Published:** 2023-01-18

**Authors:** Makoto Yanagisawa, Simon D. X. Chuong

**Affiliations:** Department of Biology, University of Waterloo, Waterloo, ON N2L 3G1, Canada

**Keywords:** autophagy, autophagosome, autophagic vacuole, chloroplast envelopes, stromules

## Abstract

Autophagy is a degradation process of cytoplasmic components that is conserved in eukaryotes. One of the hallmark features of autophagy is the formation of double-membrane structures known as autophagosomes, which enclose cytoplasmic content destined for degradation. Although the membrane source for the formation of autophagosomes remains to be determined, recent studies indicate the involvement of various organelles in autophagosome biogenesis. In this study, we examined the autophagy process in *Bienertia sinuspersici*: one of four terrestrial plants capable of performing C_4_ photosynthesis in a single cell (single-cell C_4_ species). We demonstrated that narrow tubules (stromule-like structures) 30–50 nm in diameter appear to extend from chloroplasts to form the membrane-bound structures (autophagosomes or autophagy-related structures) in chlorenchyma cells of *B. sinuspersici* during senescence and under oxidative stress. Immunoelectron microscopic analysis revealed the localization of stromal proteins to the stromule-like structures, sequestering portions of the cytoplasm in chlorenchyma cells of oxidative stress-treated leaves of *B. sinuspersici* and *Arabidopsis thaliana*. Moreover, the fluorescent marker for autophagosomes GFP-ATG8, colocalized with the autophagic vacuole maker neutral red in punctate structures in close proximity to the chloroplasts of cells under oxidative stress conditions. Together our results implicate a role for chloroplast envelopes in the autophagy process induced during senescence or under certain stress conditions in plants.

## 1. Introduction

Autophagy is a process responsible for the degradation of cytosolic and organellar materials for nutrient recycling and the removal of undesirable components in eukaryotes. Although there are several types of autophagy, two main autophagic pathways have been described in plants: microautophagy and macroautophagy [1,2]. Microautophagy involves the direct engulfing of cytoplasmic contents by the invagination of the tonoplast, followed by their release inside the vacuole. Macroautophagy is a process where a portion of cytoplasm, including organelles, is sequestered into a double- or multi-membrane structure called an autophagosome. Subsequently, the autophagosome is transported into lysosome in mammals or vacuole in yeast and plants for hydrolysis and degradation by proteases. Finally, the degraded products, such as amino acids, are reallocated and recycled [for plant autophagy, see reviews: [3,4,5,6,7]. In plants, autophagy is induced under nutrient-limiting conditions such as carbon and nitrogen starvation [8,9,10,11], during developmental events such as senescence [12,13,14], and in response to oxidative stress and pathogen attacks [15,16,17]. Autophagy observed under normal growth conditions has been suggested to serve as a housekeeping role [18,19].

Evidence connecting the relationship between autophagosome and autophagic vacuole biogenesis has been suggested in previous studies. For example, provacuoles formed tubule structures sequestering portions of cytoplasm, which eventually led to the formation of autophagic vacuoles that have been documented in the root meristematic cells of *Euphorbia characias* [20]. In a study using tobacco miniprotoplasts in which central vacuoles were removed from protoplasts, it was shown that cytoplasmic materials in newly formed vacuoles and in the presence of the cysteine protease inhibitor E-64d, indicating the participation of autophagy concomitant with the biogenesis of new vacuoles [21]. However, autophagy was not inhibited by macroautophagy inhibitors such as 3-Methyladenine, wortmannin, or LY294002, and thus the authors proposed that different types of autophagy could occur in miniprotoplasts. Similarly, lytic vacuoles have been observed specifically in chloroplast-containing cells of senescing *Arabidopsis* leaves and termed senescence-associated vacuoles (SAV) [22,23]. Moreover, their presence in mesophyll protoplasts of the *atapg7-1 Arabidopsis* mutant indicated that the biogenesis of the SAV was independent of the ATG-dependent autophagic pathway.

Although autophagy has been discovered for more than 40 years, the origin of autophagosomes and the membrane source appears to involve numerous organelles. In yeast, the preautophagosomal structure containing the ATG proteins played a central role in autophagosome formation [19,24,25,26]. Several lines of research supported that the endoplasmic reticulum (ER), or the trans-Golgi network was responsible for autophagosome generation [2,27,28]. Studies using electron 3D tomography showed interconnections between ER and premature-autophagosome membranes in mammalian cells [29,30]. Hailey et al. reported that the outer membrane of mitochondria was utilized for autophagosome biogenesis during starvation [31]. Similarly, the involvement of the plasma membrane in the formation of autophagasome has been suggested by the recruitment of Atg16L1 to the plasma membrane, autophagosomes, and clathrin-coated structures [32]. These findings implicate that various organelles may contribute to the origin of autophagosome membranes depending on cell types and situations. Although the molecular machinery involved in autophagy is conserved among eukaryotes, including mammals, yeasts, and plants [19], it is also possible that different organisms have developed novel pathways for autophagosome formation using different organelles as a source of the membrane.

Plastids are ubiquitous organelles specifically found in photoautotrophic organisms. Plastids have been known to form membrane extensions and protrusions devoid of thylakoid membranes, and tubule structures are named “stromules” because they are stroma-filled tubules [33]. Electron microscopy revealed that stromules are double-membrane structures with both outer and inner envelopes of plastids [34,35,36]. While stromules are more abundant in plastids of non-photosynthetic tissues, they are also detected in plastids of a variety of cell types and plant species [37,38]. Moreover, the close association of stromules with other organelles such as ER, mitochondria, and the nucleus has been observed in *Arabidopsis thaliana, Nicotiana tabacum,* and *Nicotiana benthamiana* [37,39,40,41]. Although the biological function of stromules remains elusive, the flow of proteins between stromules and other compartments in close proximity has been directly demonstrated [37,42,43,44,45]. However, transporting materials between plastids may not be the major and only function of stromules because the majority of plastids are not connected by stromules [38,40,45]. Regardless, the dynamic interactions between stromules and these organelles implicate that they likely have multiple roles in a variety of cellular processes.

*Bienertia sinuspersici* represents one of four terrestrial plants previously shown to perform C_4_ photosynthesis in a single cell [46,47]. The *Bienertia* species carry out C_4_ photosynthesis without requiring the traditional Kranz anatomy. In this single-cell C_4_ system, the C_4_ pathway is achieved by the spatial compartmentalization of dimorphic chloroplasts, mitochondria and peroxisomes, and photosynthetic enzymes in two distinct domains of the cytoplasm: the peripheral cytoplasmic compartment (PCC) and the central cytoplasmic compartment (CCC) [47,48]. The PCC replaces the mesophyll cell, whereas the CCC performs the function of the bundle sheath cells. C_3_ plants evolved an important mechanism known as the chloroplast photorelocation movement to protect plants from light stress [49]. In contrast, the dimorphic chloroplasts appeared to be permanently maintained in their respective cytoplasmic subdomains, the PCC and CCC, respectively, by an elaborate cytoskeletal network with limited mobility [48]. Therefore, chloroplasts in the single-cell C_4_ system are likely to be more susceptible to oxidative damage. In this study, we investigated the role of stromules in the biogenesis of autophagy-related structures during chloroplast degradation in chlorenchyma cells of *Bienertia sinuspersici* and *Arabidopsis thaliana* while undergoing senescence or under oxidative stress. It was demonstrated that chloroplast-derived stromules appeared to serve as a source of the membranes of autophagy-related structures. In addition, the in vivo localization of autophagosomes and autophagic vacuoles was illustrated by green fluorescent protein (GFP)-ATG8 fusion and neutral red (NR), respectively. The possible role of chloroplast envelopes in autophagic vacuole formation was also discussed.

## 2. Results

### 2.1. Autophagy in Stress-Induced Chlorenchyma Cells of Bienertia sinuspersici

In *Arabidopsis* seedlings, autophagy was induced by reactive oxygen species (ROS) [16]. In order to determine whether this would occur in mature leaves, isolated chlorenchyma cells of *B. sinuspersici* were treated with 20 mM H_2_O_2_ for up to 2 h and observed using transmission electron microscopy (TEM) (Figure 1). The central cytoplasmic compartment (CCC) appeared to be unaffected under low-resolution images, whereas some peripheral cytoplasmic compartment (PCC) chloroplasts dilated, while others maintained their normal oblong shape (Figure 1A). However, at a higher resolution, structures that resembled autophagosomes and autophagic vacuoles were observed in addition to chloroplasts and mitochondria in the CCC (Figure 1B). The autophagic-like bodies were mainly oval in shape with diameters ranging from 0.4 to 2.5 μm and were occasionally found in the central vacuole in some cells (Figure 1C,D). In autophagy-related structures, cytoplasmic components, including mitochondria, were found to be surrounded by multiple membranes in both the CCC and PCC (Figure 1E,F). Autophagic vacuoles, which were electron transparent, were also observed in the CCC and PCC and appeared to degrade cellular components within the structures (Figure 1G,H). However, the autophagy-related structures were rarely observed in healthy mature untreated chlorenchyma cells (Figure 1I–K). These results indicated that autophagy was induced by H_2_O_2_ in the chlorenchyma cells of mature *B. sinuspersici* leaves.

### 2.2. Contribution of Stromules in Autophagy in Chlorenchyma Cells of Bienertia sinuspersici

It has been reported that chloroplasts change their morphology under some stress conditions. Holzinger et al. described that chloroplast protrusions appeared at high temperatures (35–45 °C) while these structures were not detected at low temperatures 5–15 °C in *Arabidopsis* mesophyll chloroplasts [39]. To observe such changes under oxidative stress, stromal proteins such as ribulose 1,5-bisphosphate carboxylase/oxygenase (Rubisco) large subunit (RbcL) or pyruvate Pi dikinase (PPDK) were immunolabelled in chlorenchyma cells and observed using TEM (Figure 2). In the CCC of the control cells, the dilation and elongation of chloroplast envelopes containing stromal proteins were observed (Figure 2A,B). Small vesicles with a diameter in the range of 90–240 nm containing stromal proteins were detected mostly in the PCC and the peripheral region of the CCC along the central vacuole (Figure 2C). Only stroma-containing vehicles (SCVs) with spherical shapes were chosen and measured to prevent the inclusion of stromules. Some SCVs appeared to be releasing content to the central vacuole. On the other hand, oxidative stress-treated cells showed stromules at a higher frequency (Figure 2D,E) and varied in size and shape. Stromules were also observed in cytoplasmic strands connecting the CCC and PCC (Figure 2F). In previous studies, we have shown that RbcL mostly accumulated in CCC chloroplasts but was scarce in the PCC chloroplasts of mature *B. sinuspersici* leaves [47,48]. The densely labeled stromule by the RbcL antibody suggested that it extended from the CCC chloroplasts to the PCC through the cytoplasmic strand. Although chloroplast protrusions, stromules, and SCVs were found in chlorenchyma cells under the control conditions, they were more abundant in oxidative stress-treated cells.

We have successfully captured images showing narrow stromules with an average diameter of 36 nm (including membranes and excluding base and tip parts of the stromules), originating from chloroplasts and extending in the cytoplasm, which appears to be sequestering portions of the cytoplasm (Figure 3A–D). Careful observation revealed that labeling for stromal proteins was detected between membranes of autophagy-related structures in both the CCC and PCC of *B. sinuspersici* chlorenchyma cells (Figure 3E–G). The diameter of the stromal part of stromules was 10–20 nm, with some exceptions (Figure 4A,B). Although autophagy-related structures are generally surrounded by two apparent membranes, some structures are enclosed by multiple membranes (Figure 4C,D).

The contents of autophagy-related structures surrounded by stromules seemed to vary based on their appearance. To identify the contents, immunolabeling experiments using various antibodies were performed (Figure 5). For example, labeling for catalase was found inside the autophagy-related structure, indicating that a peroxisome was enclosed by membranes (Figure 5A). The RbcL antibody labeled an SCV and its surrounding stromules (Figure 5B). In addition, labeling for the cytosolic marker phosphoenolpyruvate carboxylase revealed that cytosolic content, along with another organelle, was also captured by autophagy-related structures (Figure 5C,D). Together with the structures related to autophagy containing mitochondria shown in Figure 1E, autophagic bodies induced by the oxidative stress treatment appeared to be generated in a non-selective manner. Furthermore, autophagy-related structures were often found in close proximity to either central vacuoles or autophagic vacuoles (Figure 5).

### 2.3. Contribution of Stromules in Autophagy in Arabidopsis Mesophyll Cells

Autophagic bodies surrounded by chloroplast-derived stromules were observed in stress-induced chlorenchyma cells of mature *B. sinuspersici*. To determine whether this process is conserved in other plant species, *Arabidopsis* was used for EGFP expression and immunoelectron microscopic analyses. First, *Arabidopsis* protoplasts were transfected with RbcS-EGFP plasmid DNA by the polyethylenglycol (PEG)-mediated method [50] and treated with H_2_O_2_ to observe chloroplast morphology. The RbcS-EGFP signal was observed mostly in the chloroplasts of non-treated protoplasts, whereas it was also found in vesicles attached to or away from chlorophyll autofluorescent signals in H_2_O_2_-treated protoplasts (Figure 6). Next, *Arabidopsis* leaves were treated with H_2_O_2_, immunolabelled with an anti-RbcL antibody, and observed using TEM (Figure 7). Cytoplasmic components, including SCVs, were observed in the central vacuole of stressed cells (Figure 7A,B). Specific reactions to RbcL were also found in stromules encasing cytoplasmic regions. Some autophagic-related structures surrounded by stromules appeared to release their contents into the central vacuole (Figure 7C), while others appeared to have degraded cellular components inside of them (Figure 7D–F). The size of autophagic bodies, excluding stromules ranged between 0.4 and 1 μm in diameter. However, stromules sequestering a larger area of the cytoplasm were occasionally observed (Figure 7G). The chloroplast invagination of cytoplasmic materials was also observed in stress-treated *Arabidopsis* leaves (Figure 7H–J). The engulfed contents appeared to be degraded in autophagic vacuoles formed in chloroplasts. These results indicate that stromules and chloroplasts contributed to autophagy in the oxidative stress-induced mesophyll cells of *Arabidopsis,* similar to that observed in *B. sinuspersici.*

### 2.4. Localization of Autophagic Vacuoles

Neutral red (NR), which stains acidic compartments, is a common dye used to visualize lysosomes in mammals. NR has also been applied to plant tissues for the observation of autophagic vacuoles in *Arabidopsis* and tobacco [18,21,22,51]. To understand how autophagic vacuole is formed, chlorenchyma cells isolated from healthy, naturally senescing, and stress-induced *B. sinuspersici* leaves were stained with NR and observed under a confocal microscope (see Figure 8A). In the control cells, NR mainly accumulated around chloroplasts (Figure 8A left panel). Interestingly, the accumulation of NR in autophagy-related structures associated with PCC chloroplasts was observed in some control cells, suggesting that acidic autophagic bodies formed in the PCC under the normal growth condition (Arrows in Figure 8A left panel). Naturally senescing and stress-induced cells showed the localized accumulation of NR, often having more than one punctate structure associated with each chloroplast (Figure 8A middle and right panels). The preferential accumulation of NR along chloroplasts in all types of cells suggested a potential relationship between the formation of autophagic vacuole and chloroplasts. The electron micrograph of the CCC in the H_2_O_2_-treated cell further supported this idea, showing autophagic vacuole formation on chloroplast envelopes and autophagosome membranes possibly derived from chloroplasts (Figure 8B). In addition, similar patterns in the autophagic vacuole formation between naturally senescing and H_2_O_2_-treated chlorenchyma cells indicated that cell death was induced by oxidative stress. Furthermore, partially degraded cytoplasmic components, including the mitochondria and stromules in autophagic vacuoles, suggested that these newly formed vacuoles contained hydrolase activity (Figure 8C).

### 2.5. Autophagy-Related Structures and Autophagic Vacuole Formation in Arabidopsis Mesophyll Protoplasts

Autophagy-related structures and autophagic vacuoles were often observed in close proximity. To investigate the relationship between these two autophagic organelles under oxidative stress conditions, the localization of an autophagosome marker protein ATG8 and autophagic vacuole marker NR were analyzed. In the control non-stressed protoplasts, such as EGFP-ATG8, were expressed in the cytosol and nucleus, whereas they accumulated in the vesicles of H_2_O_2_-treated protoplasts (Figure 9A). NR fluorescence was detected around chloroplasts in the control protoplasts as punctate structures in H_2_O_2_-treated protoplasts (Figure 9B). When EGFP-ATG8 expressing protoplasts were stained with NR to compare the localization of the fluorescent signals, some signals were independent of each other, although the other signals in the cytosol and punctate structures were overlapped in H_2_O_2_-treated protoplasts (Figure 9C). The distinct signals of EGFP and NR indicated that the formation of autophagosomes and autophagic vacuoles was derived from the independent path, whereas the overlapped signals in the punctate structures suggested the conversion from autophagosomes to autophagic vacuoles. Moreover, in mesophyll cells of stress-treated leaves, electron micrographs of autophagosomes and autophagic vacuoles with different morphologies and contents were detected as further supporting the potential autophagosome/autophagic vacuole transformation (Figure 10A–D). Similarly, in mesophyll cells of senescing Arabidopsis leaves, stromule-bounded autophagic structures were also frequently observed protruding from the edge of chloroplasts (Figure 10E,F).

## 3. Discussion

Accumulating evidence on the dynamics of stromules and participation of autophagy in the degradation of chloroplastic proteins led us to hypothesize on the plant-specific mechanism in autophagy-related structure biogenesis. Previous studies suggested that the degradation of chloroplastic proteins occurred through autophagic organelles [14,22,52,53,54,55,56]. Although these autophagic organelles have been observed with isolation membranes, the identity of the membrane origin remained undetermined in plants. Here, we investigated the localization of stromal proteins in leaf mesophyll cells under oxidative stress. Our data showed the involvement of stromules in autophagic-related structure formation. In addition, the localization analysis of autophagic vacuoles suggested the role of chloroplast envelopes in autophagic vacuole formation.

Under oxidative stress, numerous stromules and SCVs were induced in the chlorenchyma cells of *B. sinuspersici* (Figure 2D–F). In this study, the size of the stromules was 0.1–0.3 μm in diameter, which is somewhat narrower than typical stromules with a diameter ranging from 0.3 to 0.8 μm as determined in other studies [57,58]. In addition, stromules with an average diameter of 36 nm were occasionally observed (Figure 3A–D). We observed the labeling for stromal proteins in a 10–20 nm wide gap between membranes with a white appearance indicating a stromal portion of stromules. Thin stromules with a diameter of less than 100 nm have also been reported in tomatoes, *Arabidopsis,* and soybeans [36,59,60]. Our immuno-EM analysis revealed thin stromules surrounding portions of cytoplasmic components and forming autophagy-related structures, which are often associated with chloroplasts in both *B. sinuspersici* and *Arabidopsis* (Figure 3E–G and Figure 7). Ishida et al. found that a strong GFP-ATG8 signal in chloroplast protrusions co-localized with stromal DsRed signals in *Arabidopsis* leaves, which supports our observation [54]. Moreover, similar observations on the enclosure of intercellular components by stromules have been made in previous studies, including those performed more than 40 years ago, in tobacco, tomato, *Deschampsia antarctica*, and green algae suggesting that the participation of stromules in autophagy is conserved in photosynthetic organisms [61,62,63,64].

The stromule-bounded autophagy-related structure is multilamellar and composed of a central body part containing cytoplasmic components and a stromal part between the membranes of a stromule. Chiba et al. demonstrated that Rubisco-containing bodies (RCBs) with double membranes were further surrounded by other membranes, consistent with our data showing that SCVs were confined by stromules [(Figure 5B) 53]. Furthermore, autophagy-related structures with more than two apparent membranes were often found in *B. sinuspersici* indicating that multilamellar autophagic bodies might be a common characteristic in photosynthetic cells (Figure 1E, Figure 4C,D and Figure 5A). Autophagy is generally defined as a non-selective process of degrading cellular materials. However, selective autophagy degrading certain organelles has been studied and named based on the corresponding organelle subjected to degradation: for instance, pexophagy for peroxisomes [65,66], mitophagy for mitochondria [67], and chlorophagy for chloroplasts [56,68,69]. These authors demonstrated the digestion of chloroplastic proteins via RCBs in *Arabidopsis* during avirulent *AvrRps4* infection as well as in starchless mutants while being suppressed in a carbon-rich condition and in the starch-excess mutants. Whole chloroplast degradation via autophagy has also been observed in photodamaged and senescence-induced *Arabidopsis* leaves [14,56,70]. Although the degradation pathways for peroxisomes and mitochondria have not been understood in plants, the results from this study demonstrate that autophagosomes formed under oxidative stress contained various intercellular components, including peroxisomes, mitochondria, SCVs, and cytosol (Figure 1E and Figure 5). These data suggested that oxidative stress-induced autophagy is responsible for the non-selective degradation of cytoplasmic components and also indicated that the degradation of peroxisomes and mitochondria occurred, at least partially, via the autophagosome-mediated pathway in mesophyll cells. In addition, SCVs were also found in the non-stress-treated cells of *B. sinuspersici* (Figure 2C). This is in agreement with the findings provided by Prins et al. describing that Rubisco-containing vesicles were observed in mature tobacco leaves under stress and optimal conditions as well as in young leaves [55]. Regardless, whether this pathway is specifically regulated by nutrient conditions remains undetermined in this study; the chloroplastic protein degradation through the autophagic pathway might regularly occur in a normal growth condition.

In mammals, autophagosome fuses with a lysosome to form an autolysosome where the degradation of sequestered contents takes place [71]. Vacuoles are analogous to animal lysosomes with lytic activity in yeast and plants. In plants, most of the volume of the mature cell is occupied by a large central vacuole which has multiple functions, including the maintenance of turgor pressure and storage of metabolites as well as the digestion of cellular constituents [72]. In addition to a central vacuole, the formation of autophagic vacuoles has been observed in various plant tissues and cells [20,21,22,64,73,74,75]. In *B. sinuspersici* chlorenchyma cells, the CCC is composed of massive cytosol and numerous organelles packed together and forming a ball-like structure surrounded by a central vacuole [47,48,76]. After the formation of an autophagy-related structure, it appeared to be transported into the central vacuole only, where the distance of two organelles was close enough, such as in the PCC and the peripheral region of CCC (Figure 5A,D). On the other hand, an autophagic body formed deep inside of the CCC seemed to be engulfed by or transformed into autophagic vacuoles, although it is necessary to note that these vacuoles were also observed in the PCC (Figure 1H and Figure 4C,D). Findings from the present study suggested the participation of two autophagic pathways: one involved in the transportation of cellular materials into the lumen of the central vacuole, while another utilized autophagic vacuoles synthesized de novo. However, it is still possible that autophagic vacuoles could also participate in the first pathway. After the formation of an autophagy-related structure near the central vacuole, autophagic vacuoles generated in close proximity to the autophagosomes could fuse with the central vacuole making autophagosomes protrude into the vacuolar lumen to facilitate a subsequent release (Figure 5A,B,D). In the second pathway, there seem to be sequential events indicating the transformation of an autophagic body into an autophagic vacuole: first, stromules sequester cytoplasmic materials generating an autophagy-related structure (Figure 10A,E,F); next, autophagic body membranes start deteriorating (Figure 10B); third, the cellular materials are digested in the lumen of the autophagic vacuole (Figure 10C); finally, all contents including internal membranes are degraded leaving an outermost boundary membrane (Figure 10D).

The co-localization study of EGFP-ATG8 and NR signals in *Arabidopsis* protoplasts also suggests two types of autophagic organelles that are partially overlapped in punctate structures under oxidative stress (Figure 7). In agreement with this observation, numerous studies have demonstrated the requirement of ATG8a for autophagosome formation in *Arabidopsis* during nutrient stress and senescence [77,78,79]. Interestingly, the upregulation of autophagy-related genes (ATG8 and ATG12), which is reported to be involved in distinct pathways, in the formation of autophagosome was observed in drought-stressed *Caragana korshinskii* leaves: a leguminous species that thrives in arid and semi-arid habitats [80]. Moreover, the high level of NR accumulation in autophagy-related structures associated with PCC chloroplasts in non-stressed *B. sinuspersici* cells indicated that the single-cell C_4_ species might adopt a similar drought-tolerant mechanism to survive in extreme saline and arid environment (Figure 8A left panel). Similar events in the autophagic vacuole formation have also been observed in non-photosynthetic cells, including *Arabidopsis* suspension-cultured cells [74], root meristematic cells in *Euphobia* [20], and tobacco [81]. Zheng and Stahelin have described that the membrane extension originated from protein storage vacuoles enclosed in an area of cytoplasm forming multilamellar autophagosomes in the inner cortex and vascular cylinder cells [81]. The luminal space of the extensive membrane differentiated into pre-lytic vacuoles by ‘reinflation’, which expanded and engulfed an entire autophagosome, and eventually, the whole domain was transformed into a lytic vacuole. The similarities in various cell types for different conditions suggest the plasticity of the autophagosome and autophagic vacuole biogenesis and supplied membranes from different organelles.

The generation of autophagic vacuoles has been indicated to be independent of, at least, early steps of macroautophagy [21,22]. Autophagic vacuoles were electron-translucent and occasionally contained partially degraded cellular materials (Figure 1H, Figure 2E, Figure 7I, Figure 8C and Figure 10C,D). In vivo localization analysis showed that the majority of NR were accumulated along with chloroplasts regardless of the cell conditions (Figure 8A). In agreement with this, the generation of autophagic vacuole appeared to initiate along with chloroplast envelopes, including those on stromules (Figure 1E,G, Figure 2D and Figure 8B). These observations implicated the involvement of chloroplast envelopes in the autophagic vacuole formation. Moreover, the contribution of chloroplast envelopes without an extension in the autophagic vacuole formation indicated that autophagosome formation is not essential for the generation of autophagic vacuoles. This is further supported by chloroplast invaginations of cellular components followed by the on-site generation of autophagic vacuoles without forming autophagosomes (Figure 7H–J). While our findings have established the close proximity between stromules and autophagic bodies, future experiments will examine how they are directly involved in the autophagy pathway.

Since the discovery of *B. sinuspersici* as one of four terrestrial single-cell C_4_ species, a number of studies have focused on how this novel photosynthetic system develops [47,48,76,82,83,84]. However, as shown here, *B. sinuspersici* also serves as a suitable model species for the autophagy study in mesophyll cells. Unlike mature photosynthetic cells in typical plant species, in which most of the cell space is occupied by a large central vacuole and the cytoplasm is limited to the peripheral, *B. sinuspersici* chlorenchyma cells contain a large cytoplasmic area in the CCC (Figure 1A,B) [47,48]. This is advantageous, especially when ultra-thin sections are made for the TEM analysis. In addition, the compartmentation of organelles can be exploited for the study of the selective degradation of particular organelles. For instance, mitochondria are exclusively located in the CCC, whereas peroxisomes are found in both compartments (Figure 1B and Figure 5A,D). We have also established a transient gene expression protocol in *B. sinuspersici* chlorenchyma protoplasts with comparable efficiency to those previously reported for *Arabidopsis* [50,51]. Although most genes remain unidentified, this protocol would make it feasible to analyze the localization of autophagy-related proteins such as ATGs.

Here, we have provided data supporting the role of stromules in the generation of autophagy-related structures by sequestering cellular components, including mitochondria, peroxisomes, and cytosol of photosynthetic cells. Recently, a close association between stromules and ER has been established in *Catharanthus roseus* [85]. The authors documented the subcellular localization of enzymes involved in the synthesis of monoterpene indole alkaloids (MIA) in stromules and postulated a potential role for these structures in the exchange of biosynthetic intermediates to the ER where the subsequent reactions of the pathway occur [85]. Together with our data, it is possible to speculate that ER might participate in the autophagosome formation by guiding membrane extensions from various organelles to sequester cytoplasmic compartments and by supplying lipids to the extensions. Whether plastids in non-photosynthetic cells are involved in autophagosome biogenesis needs to be determined. Nevertheless, plants appear to have established a novel strategy to generate autophagic organelles by supplying chloroplast envelopes as a source of membranes. We believe that these findings offer a new perspective on autophagy in plants.

## 4. Materials and Methods

### 4.1. Plant Materials and Growth Conditions

*Bienertia sinuspersici* was grown as previously described by Lung et al. [50]. Three- to four-month-old plants were used for all experiments except senescing leaves, which were obtained from plants grown for more than ten months. *Arabidopsis thaliana* ecotype Columbia was grown in soil (1:1 Sunshine mix of LG3 and LC1) in chambers at 120 µmol m^−2^ s^−1^ irradiance at 21 °C with a photoperiod of 16/8 h, light and dark, respectively.

### 4.2. Electron Microscopy

Chlorenchyma cells of *B. sinuspersici* leaves were isolated, as described previously [50]. Isolated *Bienertia* chlorenchyma cells and *Arabidopsis* leaves, which were sectioned into small pieces, were transferred into a cell stabilizing solution containing 25 mM HEPES-KOH (pH 6.5), 5 mM KCl, 1 mM CaCl_2,_ and 1 M mannitol or 0.3 M mannitol, respectively. Samples were fixed, embedded in LR white resin (Polysciences Inc., Warrington, PA, USA), and sectioned as described previously [86]. Sections were stained with uranyl acetate and lead citrate.

Immunolabeling was performed as described previously [86] by using anti-Rubisco large subunit antibodies (Agrisera, Vänäs, Sweden, 1:1000), anti-*Zea mays* PPDK antibodies (courtesy of Chris Chastain, 1:1000), anti-catalase antibodies (courtesy of Robert Mullen, 1:20), anti-PEPC antibodies (Agrisera, Vänäs, Sweden, 1:1000), or anti-GDC antibodies (Agrisera, Vänäs, Sweden, 1:1000). As controls and all immunolabelling experiments were performed without the primary antibody or with preimmune rabbit sera. Images were taken using a Philips CM-10 transmission electron microscope (FEI Company, Hillsboro, OR, USA) at an accelerating voltage of 60 kV. All measurements of diameter were measured in at least ten micrographs of the samples from three independent experiments using Image J (http://rsbweb.nih.gov/ij/, accessed on 30 June 2011).

### 4.3. Plasmid Construction and Protoplast Transfection

The EGFP-ATG8 and RbcS-EGFP fusion constructs were generated by subcloning corresponding DNA fragments into the pSAT6-35S::EGFP-C1 or pSAT6-35S::EGFP-N1 vectors [87], respectively. Briefly, the coding sequence of *Arabidopsis* ATG8a was amplified by using forward (5′-AGA GTC GAC ATG GCT AAG AGT TCC TTC AAG AT-3′) and reverse (5′-AGA GGA TCC TCA AGC AAC GGT AAG AGA TCC-3′) primers designed according to Yoshimoto et al. and fused into the *SalI-BamHI* sites of pSAT6-35S::EGFP-C1 [88]. The cDNA encoding *Arbidopsis* RbcS was amplified by using forward (5′-AGA CCA TGG CTT CCT CTA TGC TCT CTT-3′) and reverse (5′-ATA GGA TCC ACC GGT GAA GCT TGG TGG-3′) primers and fused into the NcoI-BamHI sites of pSAT6-35S::EGFP-N1.

Arabidopsis protoplast transfection was performed according to Yoo et al. [51].

### 4.4. Confocal Laser Scanning Microscopy

Isolated chlorenchyma cells of *B. sinuspersici*, wildtype, and EGFP-ATG8 transfected *Arabidopsis* protoplasts were stained with 0.01 or 0.005% (*w*/*v*) neutral red, respectively, for 30 min. Samples were washed and fixed in a fixative solution containing 0.3 M mannitol, 50 mM PIPES (pH 7.2), 1.25% (*v*/*v*) glutaraldehyde, and 2% (*v*/*v*) paraformaldehyde for 10 min. Images were taken using an Olympus FV1000 confocal laser scanning microscope (Olympus, Breiningsville, PA, USA). An EGFP signal was excited at 488 nm, and the emission was detected at 512 nm. Neutral red was excited at 577 nm, and the emission was detected at 592 nm. The excitation wavelength for chlorophyll autofluorescence was 649 nm, and the emission was 666 nm.

### 4.5. Hydrogen Peroxide Treatment

For an oxidative stress treatment, *B. sinuspersici* mature leaves and isolated chlorenchyma cells were incubated in cell-stabilizing solutions containing 20 mM hydrogen peroxide (H_2_O_2_) for 2 h at room temperature. For the treatment of *Arabidopsis*, 10 or 5 mM H_2_O_2_ was used for leaves or protoplasts, respectively. A different set of samples was also incubated in a cell stabilizing solution without H_2_O_2_ as the negative control.

## Figures and Tables

**Figure 1 plants-12-00443-f001:**
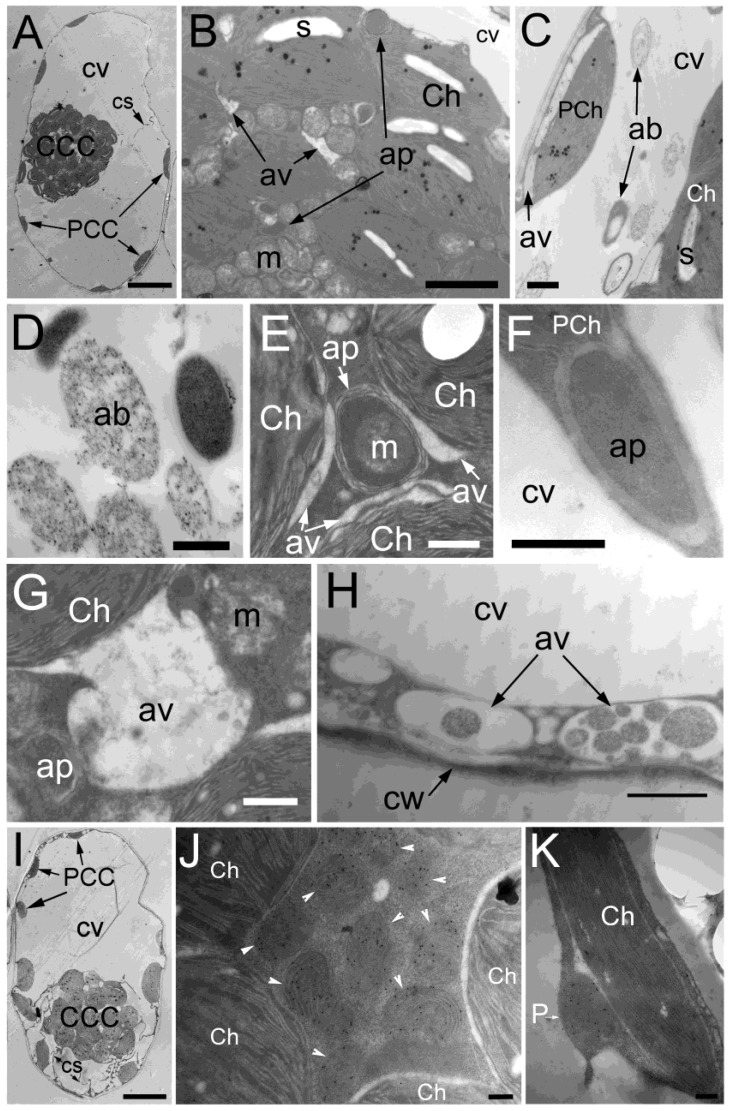
Transmission electron micrographs of *B. sinuspersici* chlorenchyma cells under oxidative stress. Mature leaves of *B. sinuspersici* were treated with 20 mM hydrogen peroxide, processed, and observed using transmission electron microscopy (TEM). (**A**) Images are an entire chlorenchyma cell treated with H_2_O_2_ for 30 min; (**B**) Various organelles in the CCC; (**C**,**D**) Autophagic bodies in the central vacuole; (**E**) An autophagy-related structure containing mitochondria surrounded by autophagic vacuoles associated with chloroplasts; (**F**) An autophagy-related structure associated with a PCC chloroplast; (**G**) Autophagic vacuole formed in close proximity to chloroplasts in the CCC; (**H**) Autophagic vacuoles in the PCC; (**I**) A chlorenchyma cell without oxidative stress treatment (**J**). The CCC of an untreated chlorenchyma cell probed with the mitochondrial marker, glycine decarboxylase; (**K**) the PCC of an untreated chlorenchyma cell probe with the peroxisomal marker, catalase. CCC, central cytoplasmic compartment; PCC, peripheral cytoplasmic compartment; cv, central vacuole; PCh—PCC chloroplast; ap—autophagy-related structure; av—autophagic vacuole; Ch—chloroplast; cs—cytoplasmic strand; s—starch grain; m—mitochondrion; ab—autophagic body; cw—cell wall; P—peroxisome; arrowheads, mitochondria. Scale bars = 10 μm in (**A**,**I**), 2 μm in (**B**), 1 μm in (**C**,**H**), 500 nm in (**D**–**G**), 100 nm in (**J**,**K**).

**Figure 2 plants-12-00443-f002:**
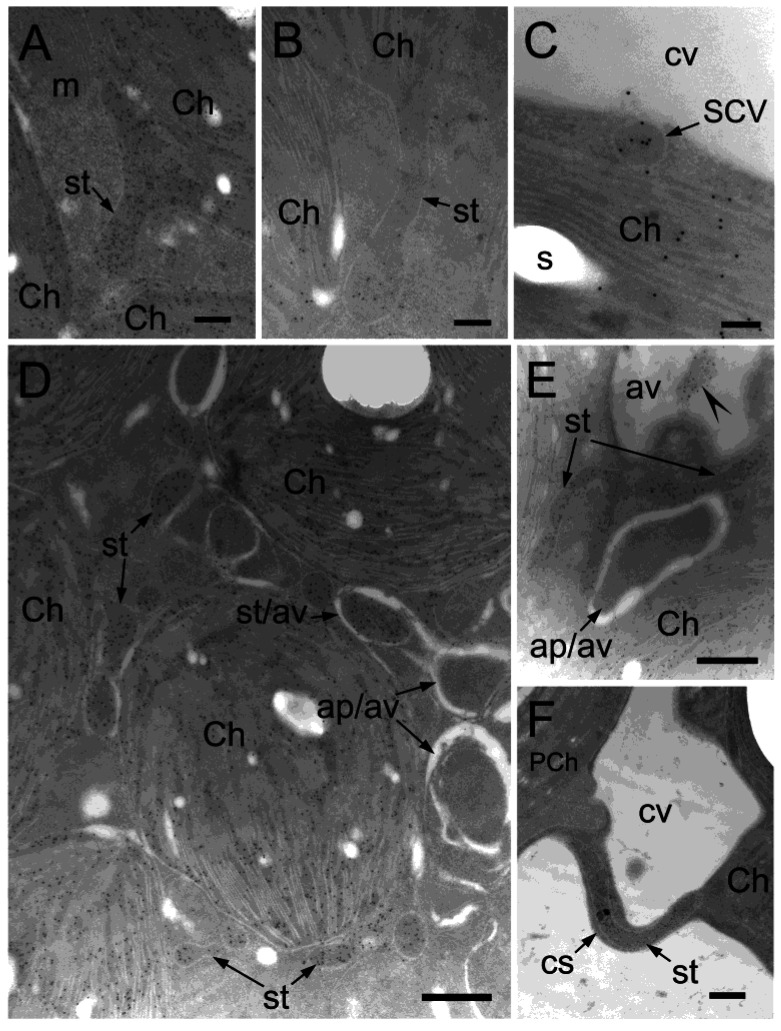
Various forms of stromules detected by immunolocalization analysis. (**A**–**C**) Cross-sections of mature *B. sinuspersici* leaves were prepared non-treated; (**D**–**F**) Treated with 20 mM hydrogen peroxide. Sections were probed with RbcL or PPDK antiserum and then gold-conjugated secondary antibody. Images are transmission electron micrographs showing the specific reaction of RbcL or PPDK antibody. (**A**,**B**) Stromules extending from chloroplasts in the CCC; (**C**) A stroma-containing vesicle (SCV) in the PCC cytosol sandwiched between a PCC chloroplast and the central vacuole appeared to be releasing its stromal content into the central vacuole; (**D**) Numerous oval-shaped stromules and autophagy-related structures in the CCC are detected, many of which are surrounded by autophagic vacuoles; (**E**) Branched stromule, autophagic vacuole degrading stromule, and autophagosome surrounded by autophagic vacuoles are observed; (**F**) A stromule densely labeled by gold particles and specifically bound to RbcL is detected in a cytoplasmic strand connecting the CCC and the PCC. Ch, chloroplast; st—stromule; m—mitochondrion; cv—central vacuole; ap—autophagy-related structure; av—autophagic vacuole; CCC—central cytoplasmic compartment; PCC—peripheral cytoplasmic compartment; PCh—PCC chloroplast; cs—cytoplasmic strand. Scale bars = 200 nm in (**A**,**B**), 100 nm in (**C**), 500 nm in (**D**–**F**).

**Figure 3 plants-12-00443-f003:**
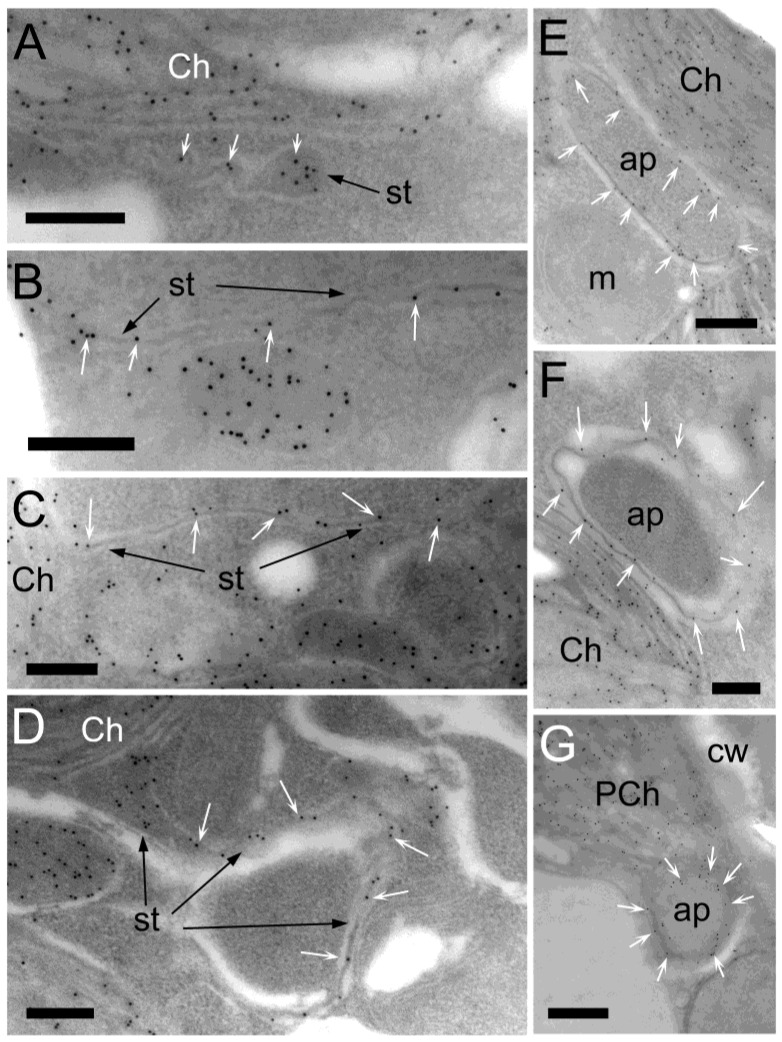
Narrow stromules surrounding autophagy-related structures. Cross-sections were prepared from *B. sinuspersici* leaves treated with oxidative stress. (**A**–**D**) Narrow stromules with an average of 36 nm in diameter extending from chloroplasts; (**C**,**D**) Some stromules appeared to be sequestering a portion of cytoplasm; (**E**,**F**) The membrane part of autophagy-related structures associated with the CCC chloroplasts consist of narrow stromules showing specific reactivity to anti-RbcL antibodies; (**G**) The membrane surrounding the autophagy-related structure in the proximity of the PCC reacted with anti-PPDK antibodies. White arrows indicate gold particles specifically bound to RbcL (**A**–**F**) or PPDK (**G**). CCC—central cytoplasmic compartment; PCC—peripheral cytoplasmic compartment; Ch—chloroplast; st—stromule; ap—autophagy-related structure; m—mitochondrion; PCh—PCC chloroplast; cw—cell wall. Scale bars = 200 nm in (**A**–**F**), 500 nm in (**G**).

**Figure 4 plants-12-00443-f004:**
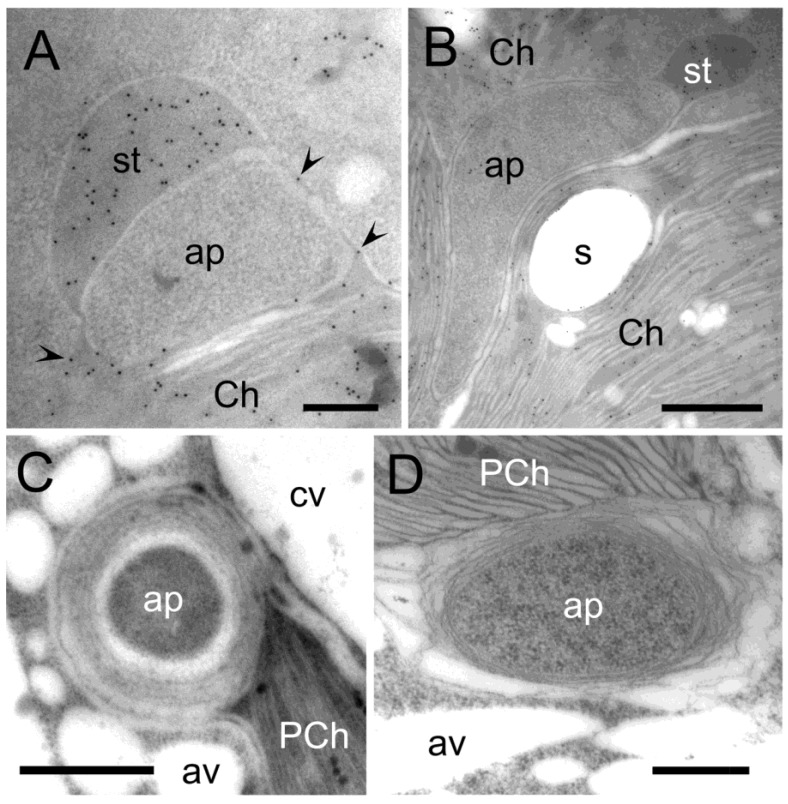
Various forms of autophagy-related structure membrane. Transmission electron micrographs of autophagic bodies with different membrane morphologies are shown. Cross-sections were prepared from mature *B. sinuspersici* leaves treated with hydrogen peroxide. (**A**,**B**) Sections were probed with RbcL antiserum and then a gold-conjugated secondary antibody. Autophagy-related structures are surrounded by stromules with narrow and thick regions detected by their specific reaction to anti-RbcL antibodies in the CCC. Arrowheads indicate gold particles. (**C**,**D**) Numerous membranes were observed on autophagy-related structures associated with PCC chloroplasts. st—stromule; ap—autophagy-related structure; Ch—chloroplast; s—starch grain; av—autophagic vacuole; CCC—central cytoplasmic compartment; PCC—peripheral cytoplasmic compartment; PCh—PCC chloroplast. Scale bars = 200 nm in (**A**), 500 nm in (**B**,**D**), 1 μm in (**C**).

**Figure 5 plants-12-00443-f005:**
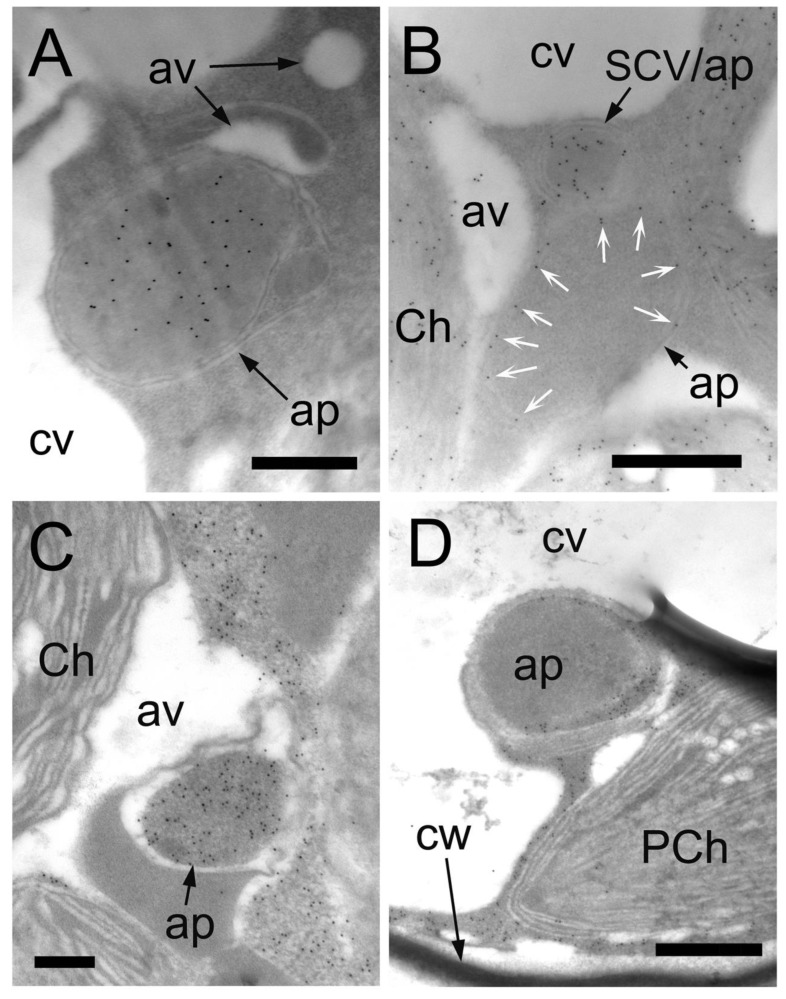
Identification of contents in various autophagy-related structures. Cross sections were prepared from mature *B. sinuspersici* leaves treated with hydrogen peroxide. Sections were probed with antiserum raised against (**A**) Catalase; (**B**) RbcL; or (**C**,**D**) PEPC as markers for peroxisome, the stroma of chloroplast, or cytosol, respectively. Sections were then probed with gold-conjugated secondary antibodies and observed using TEM. Micrographs show the specific reactivity of the antibodies against these enzymes. White arrows indicate gold particles specifically bound to RbcL in the narrow stromule surrounding a cytoplasmic portion. ap—autophagy-related structure; av—autophagic vacuole; cv—central vacuole; Ch—chloroplast; SCV—stroma-containing vesicle; PCh—PCC chloroplast; cw—cell wall. Bars = 500 nm in (**A**,**B**,**D**), 200 nm in (**C**). PCC, peripheral cytoplasmic compartment; PCh, PCC chloroplast. Scale bars = 200 nm in (**A**), 500 nm in (**B**,**D**), 1 μm in (**C**).

**Figure 6 plants-12-00443-f006:**
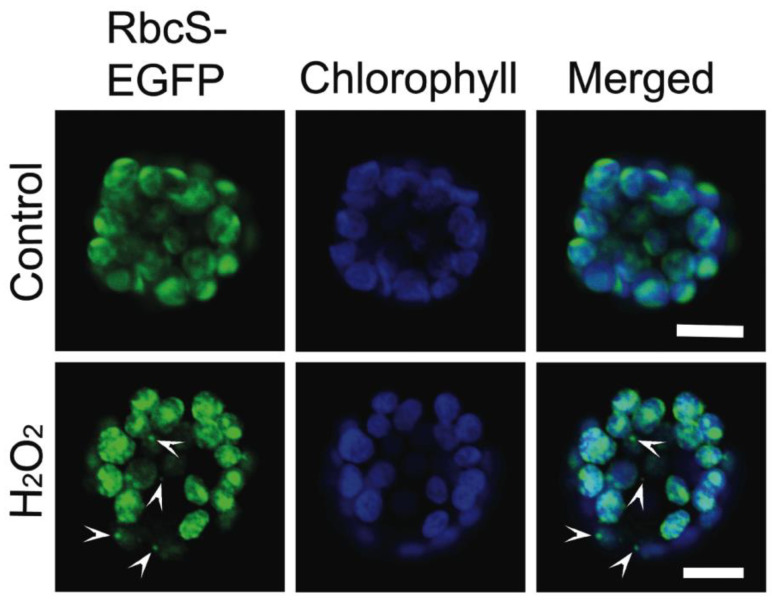
Changes in the chloroplast morphology of *Arabidopsis* mesophyll protoplasts under oxidative stress. *Arabidopsis* mesophyll protoplasts were transfected with the stroma targeting RbcS-EGFP plasmid construct, incubated with 5 mM hydrogen peroxide (H_2_O_2_) or without (Control) for two hours, and observed using confocal microscopy. Micrographs are the RbcS-EGFP fluorescent (green) and chlorophyll autofluorescent (blue) signals and merged images. Arrowheads indicate the RbcS-EGFP fluorescent signals localized in vesicles. Scale bars = 10 μm.

**Figure 7 plants-12-00443-f007:**
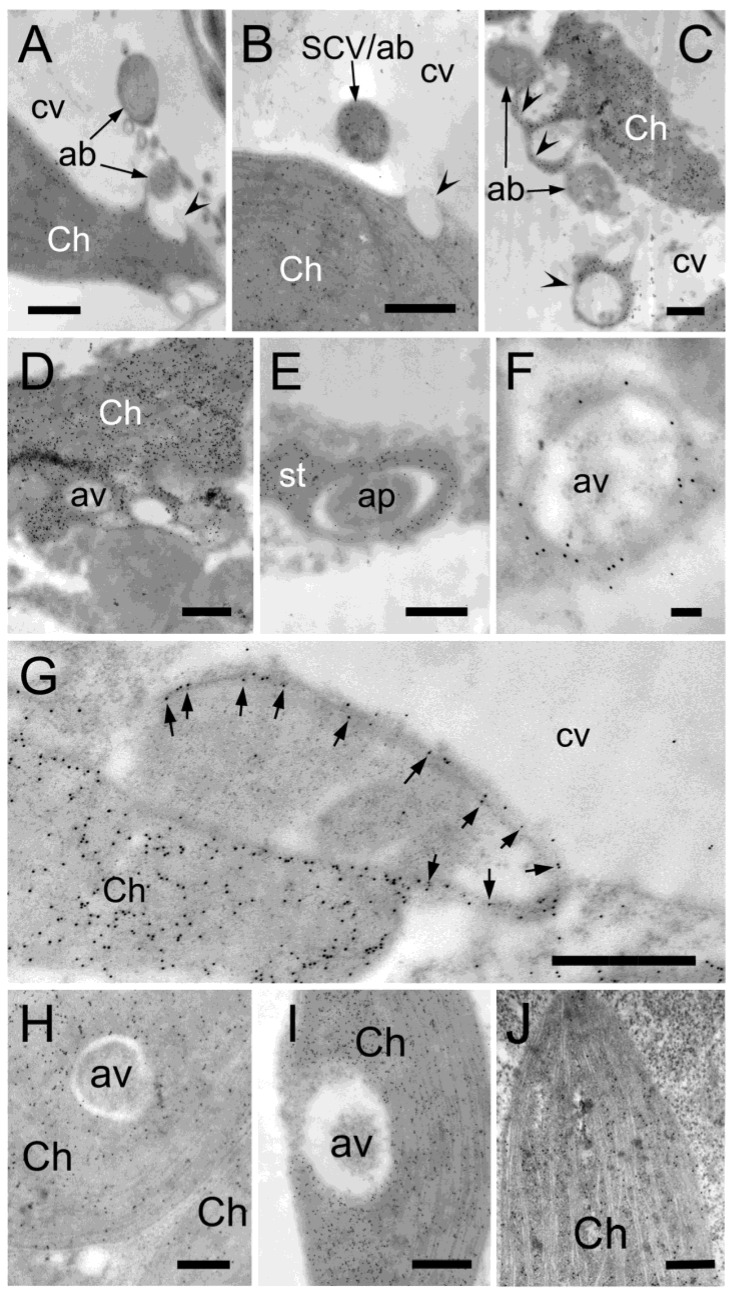
Contribution of stromules in autophagy-related structure and autophagic vacuole formation detected by immunolocalization of RbcL in H_2_O_2_-treated *Arabidopsis* leaves. Cross-sections were prepared from *Arabidopsis* leaves treated with 10 mM hydrogen peroxide for two hours. Sections were probed with RbcL antiserum and then a gold-conjugated secondary antibody. Images are transmission electron micrographs showing the specific reaction of the RbcL antibody. (**A**) Autophagic bodies are detected in the central vacuole in close proximity to the chloroplastic cavity (an arrowhead); (**B**) A stroma-containing vesicle (SCV) localized in the central vacuole is also found near the chloroplastic cavity (an arrowhead); (**C**) Autophagic bodies and ring-shaped stromules (arrowheads) are shown. The ring-shaped stromules are either attached to or away from the chloroplast; (**D**) Stromules surround multiple cytoplasmic regions and possibly generate autophagic vacuoles; (**E**) Stromules sequester a portion of cytoplasm, forming an autophagic body and an autophagic vacuole; (**F**) Autophagic vacuole is surrounded by stromules; (**G**) A stromule extending from a chloroplast sequesters a cytoplasmic area. Arrows indicate specific reactions to RbcL in the stromule. (**H**,**I**) A portion of cytoplasm is observed in the chloroplast stroma, and its partial degradation is also detected; (**J**) Cross-sections prepared from control *Arabidopsis* leaves probed with RbcL antiserum. ab—autophagic body; Ch—chloroplast; cv—central vacuole; av—autophagic vacuole; st—stromule; ap—autophagy-related structure. Scale bars = 500 nm in (**A**–**E**,**G**,**I**,**J**), 200 nm in (**H**), 100 nm in (**F**).

**Figure 8 plants-12-00443-f008:**
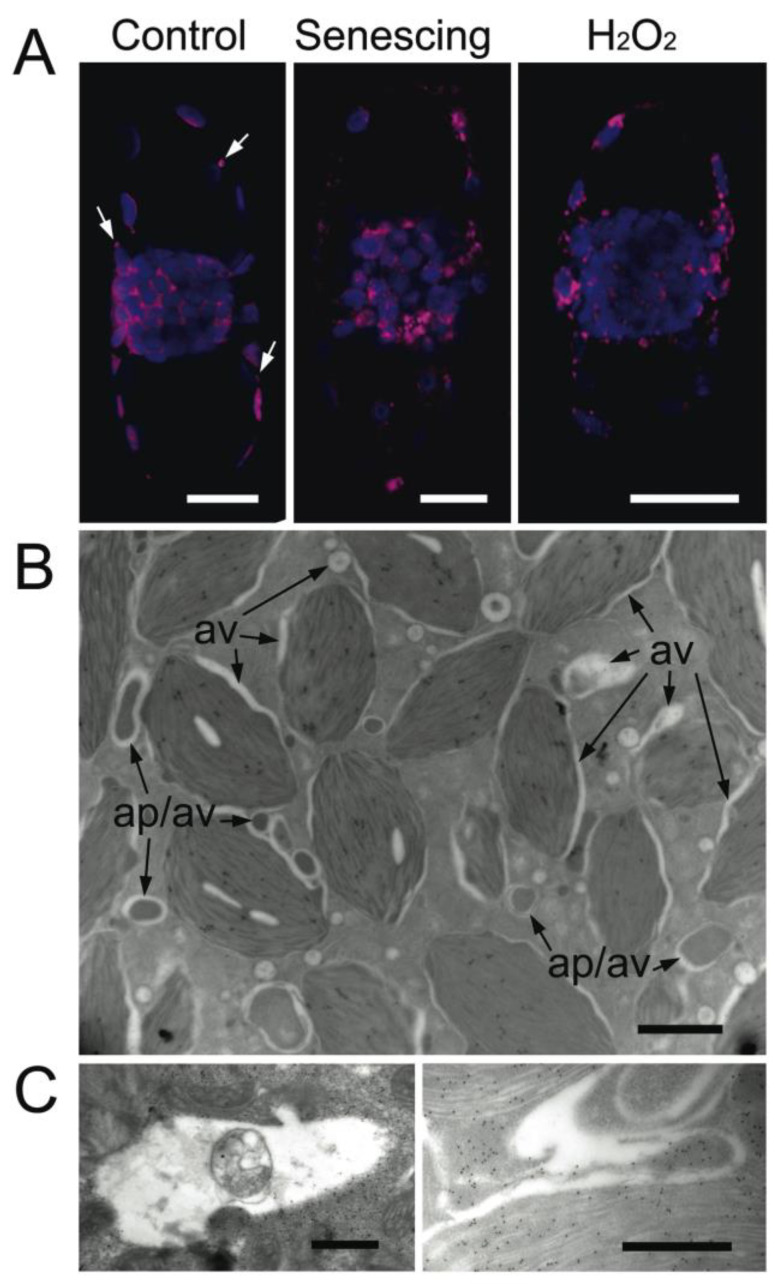
The formation, localization, and property of autophagic vacuoles. Autophagic vacuoles were observed in mature chlorenchyma cells of *B. sinuspersici* using confocal microscopy and TEM. (**A**) Chlorenchyma cells isolated from healthy, senescing, and H_2_O_2_-treated leaves were stained with the autophagic vacuole marker neutral red. White arrows indicate autophagic vacuoles associated with PCC chloroplasts in a control cell; (**B**) Sections were prepared from leaves treated with hydrogen peroxide for TEM analyses; (**C**) Autophagic vacuoles form in close proximity to chloroplasts and autophagy-related structures. Partially degraded mitochondria and stromules were detected in autophagic vacuoles. av—autophagic vacuoles; ap—autophagy-related structure. Scale bars = 10 μm in (**A**), 2 μm in (**B**), 500 nm (**C**).

**Figure 9 plants-12-00443-f009:**
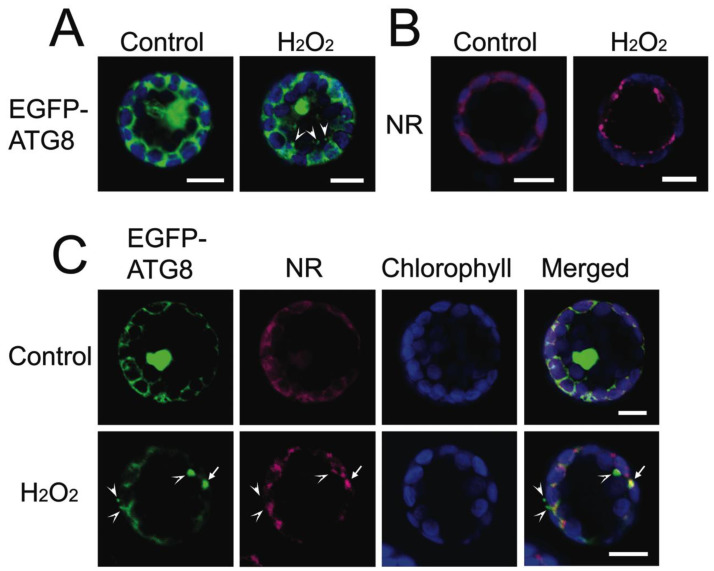
Localization of autophagosomes and autophagic vacuoles in *Arabidopsis* mesophyll protoplasts. The localization of autophagosomes and autophagic vacuoles was detected by using a transient expression of the EGFP-ATG8 plasmid construct and the fluorescent dye neutral red (NR) in *Arabidopsis* mesophyll protoplasts. Protoplasts were incubated in the presence of 5 mM of hydrogen peroxide (H_2_O_2_) or absence (Control) for two hours. Images were taken by confocal microscopy. Protoplasts were transfected with EGFP-ATG8 plasmid DNA driven by the constitutive 35S promoter. (**A**) The EGFP-ATG8 fluorescent (green) and chlorophyll autofluorescent (blue) signals are shown; (**B**) Protoplasts were stained with NR. The NR fluorescent (magenta) and chlorophyll autofluorescent (blue) signals are shown; (**C**) The EGFP-ATG8 expressing protoplasts were stained with NR. The EGFP-ATG8 (green), NR (red), and chlorophyll (blue) fluorescent signals and merged images are shown. An arrow indicates the co-localization of EGFP-ATG8 and NR signals in a punctate structure, whereas arrowheads indicate that EGFP-ATG8 and NR signals are independently localized. Scale bars = 10 μm.

**Figure 10 plants-12-00443-f010:**
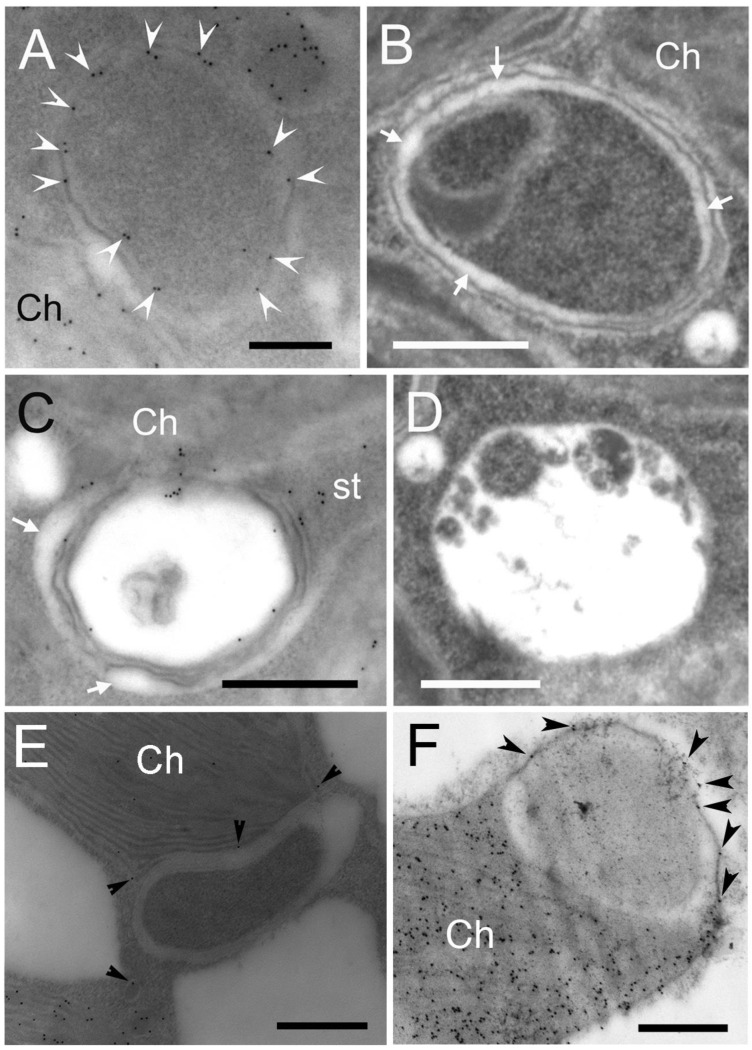
Transmission electron micrographs illustrate the transition from autophagic-related structures to autophagic vacuoles. Cross-sections were prepared from mature *Arabidopsis* leaves treated with hydrogen peroxide and senescing leaves. (**A**) TEM micrograph showing an autophagy-related structure associated with stromules detected by specific RbcL reactivity; (**B**) An autophagy-related structure with thick membranes; (**C**) A potential intermediate form of an autophagy-related structure and autophagic vacuole with partially degraded contents in the lumen although autophagic body membranes remain; (**D**) Another autophagic vacuole with partially degraded contents in the lumen, but its membranes are not visible; (**E**) Autophagic structure with cytoplasmic or stromal content protruding from a chloroplast of senescing leaves; (**F**) Stromule extending from the edge of a chloroplast forming an autophagy-related structure in senescing leaves. Arrowheads indicate RbcL-gold particles. Arrows indicate thick regions on autophagy-related structure membranes. Ch—chloroplast; st—stromule. Scale bars = 200 nm in (**A**,**E**,**F**), 500 nm in (**B**–**D**).

## Data Availability

Not applicable.

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
