# Peer review of "Chloroplast Envelopes Play a Role in the Formation of Autophagy-Related Structures in Plants"

_plants, 2023, doi:10.3390/plants12030443_

Round 1
Reviewer 1 Report
Manuscript Review
Chloroplast envelopes play a role in the formation of autophagy-related structures in plants
By Makoto Yanagisawa and Simon D.X. Chuong
Submitted to Plants (Plants-2156347)
In this paper, the authors examined the role of chloroplast envelopes in the formation of autophagy-related structures in the C4 plant Bienertia sinuspersici. They have shown that narrow tubules extend from chloroplasts and form the membrane-bound structures in chlorenchyma cells of the plant during senescence and under oxidative stress. I believe this manuscript requires proper revisions before it can be considered for publication.
Abstract
L10. A comma is needed before “which”. It applies to other places.
L11. What is the rationale of this study?
L12-14. This is a phrase but not a sentence. It needs to be modified.
L18. Is there a common name for this species? If yes, it should be added.
Introduction
L35. A comma is needed before “such as” and after “amino acids”. It applies to other places.
L44. Euphorbia; Is there a species name here?
L46-47. It is not clear and should be rewritten.
L65. “localization” needs to be used in the correct form.
L80. Both common and Latin names should be written in the first mention.
L87. It is “autophagy”.
L89. Clear hypothesis and objectives are needed.
Results
L126. It is better to reword “chloroplast changed”.
L140. connecting the CCC and PCC to what? It should be reworded.
L217. It is better to replace the first “in” with “to”.
Figure 8. The letter “D” is missing from the figure.
L312. Is “stress-treated” the right term?
Discussion
L357. It is B. sinuspersici.
Materials and Methods
L500. It is better to mention the location and time of this study.
L504. Is it “16h/8h, light and dark, respectively”?
L542. A reference needs to be cited.
L548. A reference needs to be cited.
References
L565. The journal name should be in italics. It applies to all references.
L572-573. The repeated part should be deleted.
L578. The period after “Protoplasma” should be deleted. It applies to other places (L597, 609,629,638,649,655,696,701,707,711,720,726).
L579. The period after “Acta” should be deleted. It applies to other places (L697).
L640. It is better to write “J. Exp. Bot.”.
L716. Is “Systematic Bot.” the correct abbreviated?
L742. “Versatile”; Reference 84 was not found in the text. All the references should be checked, cited vs. listed and listed vs. cited.
Reviewer 2 Report
The manuscript titled: “Chloroplast envelopes play a role in the formation of autophagy-related structures in plants” by Makoto Yanagisawa and Simon D.X. Chuong submitted to Plants, gives results on the role of chloroplast envelopes in the autophagy process during the natural senescence or of the oxygen- stress conditions in plants in two species: Arabidopsis thaliana, a well know model plants, and Bienertia sinuspersici, known by authors from their earlier investigations.
The introduction in a consisted way gives information on the role of autophagy in the natural and induced senescence as well as on the course of such process pointing out the variety of organelles contributing to the origin of autophagosome membranes which depends on cell types and organisms. The introduction gives the reason why stromules are potential autophagy-related structures. I highly appreciate the clarity of presentation of the main subject of this manuscript. It will be useful, however, to explain, in the Introduction or later in Results, to explain the specificity of the photosynthetic apparatus of Bienertia sinuspersici, which conducts a single-celled C4 photosynthesis within individual chlorenchyma cells. In my opinion the explanation of the unusual dimorphism of this C4 plant helps to understand the structure and function of two distinct cellular compartments: the central chloroplast compartment (CCC) and the peripheral chloroplast compartment (PCC), which are the important sites described by the authors in the manuscript.
The obtained results demonstrate the involvement of the stromule-like structures, extended from chloroplasts, in the formation of autophagy-related structures in chlorenchyma cells of Bienertia sinuspersici during the senescence and under oxidative stress as well as localization of stromal proteins in such structures in oxidative stress-treated leaves both of B. sinuspersici and Arabidopsis thaliana. I especially appreciate the results obtained by immunolocalization of GFP-ATG8 (a fluorescent marker for autophagosomes), in close proximity to chloroplasts under oxidative stress conditions.
Material and methods are very well and clearly described. Experimental methods especially immunolocalization both in TEM and confocal microscopy are well performed with very good quality. However, why autofluorescence of chlorophyll shown in Figs. 6 and 9 is blue and not red? As it is stated in Material and methods: “The excitation wavelength for chlorophyll autofluorescence was 649 nm and 541 the emission was at 666 nm”.
The results of B. sinuspersici chlorenchyma cells as well as these from leaves of B. sinuspersici and A. thaliana are shown under oxidative stress only, pictures of the control are not given, why? Images of the control conditions are required not only in confocal microscopy.
My remarks:
· I am curious why the authors do not give their affiliation in the manuscript, only e-mail address. I will appreciate authors affiliation.
· Referring to my earlier remark, please explain CCC and PCC compartments which are the important sites.
· Why autofluorescence of chlorophyll gives blue and not red color?
· Please add control pictures to all figures.
The manuscript is written clearly and in a comprehensive way.
Concluding:
I positively evaluate the results obtained in the manuscript titled: “Chloroplast envelopes play a role in the formation of autophagy-related structures in plants” by Makoto Yanagisawa and Simon D.X. Chuong. The authors gave the evidence that chloroplast envelopes play a role in the autophagy process during the natural senescence and autophagy induced by oxidative stress pointing the involvement of stromule-like structures as a source of membranes in such process.
After minor corrections I recommend the manuscript to be published in Plants.
